# The Novel Serine/Threonine Protein Kinase LmjF.22.0810 from *Leishmania major* May Be Involved in the Resistance to Drugs such as Paromomycin

**DOI:** 10.3390/biom9110723

**Published:** 2019-11-11

**Authors:** Andrés Vacas, Celia Fernández-Rubio, Miriam Algarabel, José Peña-Guerrero, Esther Larrea, Fabio Rocha Formiga, Alfonso T. García-Sosa, Paul A. Nguewa

**Affiliations:** 1Department of Microbiology and Parasitology, ISTUN Institute of Tropical Health, University of Navarra, IdiSNA (Navarra Institute for Health Research), Pamplona, E-31008 Navarra, Spain; 2Aggeu Magalhães Institute, Oswaldo Cruz Foundation (FIOCRUZ). Graduate Program in Applied Cellular and Molecular Biology (BCMA), University of Pernambuco (UPE), Recife/PE 50.670-420, Brazil; fabio.formiga@cpqam.fiocruz.br; 3Institute of Chemistry, University of Tartu, Ravila 14a, 54011 Tartu, Estonia; alfonso.tlatoani.garcia.sosa@ut.ee

**Keywords:** *Leishmania*, NTD, docking, molecular dynamics, drug resistance, paromomycin, kinase, treatment, LmjF.22.0810, LmJean3

## Abstract

The identification and clarification of the mechanisms of action of drugs used against leishmaniasis may improve their administration regimens and prevent the development of resistant strains. Herein, for the first time, we describe the structure of the putatively essential Ser/Thr kinase LmjF.22.0810 from *Leishmania major*. Molecular dynamics simulations were performed to assess the stability of the kinase model. The analysis of its sequence and structure revealed two druggable sites on the protein. Furthermore, in silico docking of small molecules showed that aminoglycosides preferentially bind to the phosphorylation site of the protein. Given that transgenic LmjF.22.0810-overexpressing parasites displayed less sensitivity to aminoglycosides such as paromomycin, our predicted models support the idea that the mechanism of drug resistance observed in those transgenic parasites is the tight binding of such compounds to LmjF.22.0810 associated with its overexpression. These results may be helpful to understand the complex machinery of drug response in *Leishmania*.

## 1. Introduction

In 2006, paromomycin (PMM) was introduced for the treatment of visceral leishmaniasis. Its effectiveness and low cost allowed its use as a first-line alternative in the setting of resistance to traditional antileishmanial treatments [1]. PMM belongs to the aminoglycoside (AG) antibiotic family that has been broadly used for treating Gram-negative bacterial infections [2,3]. However, the emergent resistance to these antibiotics, conferred principally by bacterial eukaryotic-like protein kinases (ePKs), has left AGs ineffective against several pathogenic strains [2,3,4]. Concerning PMM alternatives against leishmaniasis, the emergence of drug-resistant strains has already rendered antimony useless against *Leishmania* infections; furthermore, other second-line drugs such as pentamidine have reported failure rates of around 30% in India [5]. On the other hand, the cost concerns of miltefosine or lipid formulations of amphotericin B remain a considerable challenge for the control and reduction of the burden of leishmaniasis [5,6]. Therefore, PMM stands as a convincing treatment for a broader usage [1]. However, even after its approval and introduction into the clinic, its mechanisms of action still need to be further studied to understand the development of resistance in *Leishmania* [7].

Trypanosomatid signaling pathways are not likely to culminate at the level of transcription since *Trypanosoma* and *Leishmania* transcribe their genes as polycistronic units with little evidence of regulation [8]. Therefore, as a consequence of inference with other eukaryotic cells, it is expected that protein phosphorylation may be critical for parasite’s development [9]. In fact, changes in the phosphorylation pattern and expression of ePKs and phosphatases in *Leishmania* and other kinetoplastids during their life cycles are common phenomena and have been demonstrated [10,11,12,13]. Protein kinases (PKs) are key regulators of cellular signaling and have long been recognized as important drug targets in a variety of diseases (such as cancer and inflammatory and infectious diseases) [14,15,16,17]. The human kinome (518 kinases, ~2% of the human genome) is the largest “druggable” family of molecular targets in humans [18]. Kinases also represent a relatively large group of the predicted protein-coding genes from trypanosomatid parasites (approximately 2% of each genome). Protein essentiality, druggability, and the absence of orthologs in the human proteome are some of the key features that should be displayed by a potential therapeutic target [19]. Interestingly, PKs play vital roles in *Leishmania* and *Trypanosoma* such as the control of virulence and growth of parasites, allowing their consideration as potential therapeutic targets against neglected tropical diseases (NTDs) [20,21,22,23].

Several dozens of small-molecule kinase inhibitors have been either approved or studied for the treatment of various human diseases which include cancer, Alzheimer’s disease, as well as infections [20,24,25,26,27]. Nevertheless, due to the phylogenetic proximity with their hosts, targeting ePKs from eukaryotic microorganisms is more challenging than dealing with bacterial PKs [6]. Analyses conducted by Florez et al. (2010) determined phosphoproteins as the largest cluster among the 384 potential drug targets discovered in *Leishmania major* [28]. Amongst these proteins, 91 kinases were predicted to be essential for parasite’s survival and also shown to lack orthologous proteins in the human kinome. Thus, focusing on peculiarities found in PKs from eukaryotic microorganisms is essential to avoid off-target activity and cross-reactivity of inhibitors with kinases from their hosts [6,18].

Fang et al. (2013) noticed that understanding the structural details of kinase conformational changes eventually allows the rational design of innovative kinase inhibitors [29]. Moreover, the structure of the enzyme-bound antagonist complex is used in the classification of kinase inhibitors [30]. Most of the FDA-approved kinase inhibitors (type I) reversibly bind to the adenine site of the ATP pocket from kinases in their active state [14], therefore preventing the co-factor ATP from binding to the site. However, they have to compete with ATP towards binding to this region [18]. The structure of active kinases is highly conserved in all organisms; on the contrary, kinases in their inactive state are structurally more diverse and dynamic [31]. Type II and III inhibitors target kinases during their inactive state, whereas type IV inhibitors target regions outside of the ATP cleft and the phosphoacceptor site in active and inactive kinases [30,31,32]. Similarly, in recent years, increased attention has been placed on small molecules that target the protein substrate binding site [18]. Nevertheless, it is well known that designing molecules for such relatively flat surfaces is tricky. Consequently, such a challenging strategy has not yet been extensively studied [32].

Since calcium ion (Ca^2+^) is an important second messenger in the biology of trypanosomatid parasites and is implicated in their differentiation, host cell invasion, and flagellum motility [33], our study was focused on the characterization of a protein from the Ca^2+^/calmodulin-dependent group of kinases (CAMK) in *Leishmania.* This novel molecule has also been predicted to be a therapeutic target [28]. Our work represents the first report describing this gene (LmjF.22.0810). Considering the lack of orthologous sequences, we decided to name this gene “Jean3”. Moreover, due to the unavailability of an X-ray 3D structure of the protein LmjF.22.0810, we were prompted to develop for the first time a homology model for LmjF.22.0810. Various methods were then employed to validate the predicted structure before it was used to study in silico its ability to bind different pharmacological substrates, including several AGs. After cloning and overexpressing LmjF.22.0810 in *L. major,* we found that such parasites were less sensitive to PMM and geneticin (G418). These outcomes, together with the comparative analysis of the primary sequence of the protein, allow us to propose a novel mechanism of AG resistance in *Leishmania* mediated by the tight binding of AGs to the protein substrate site of LmjF.22.0810. Our results will be valuable not only to understand the complexity of drug resistance in *Leishmania* but also to design novel inhibitors against homologues of LmjF.22.0810 kinases.

## 2. Materials and Methods

### 2.1. Database Inspection and Nucleotide and Protein Sequence Alignment

Nucleotide and genomic sequence data, as well as Jean3 orthologous protein sequences, were obtained from TriTrypDB [34]. Multiple sequence alignments were prepared using the MUSCLE algorithm [35] implemented in Geneious v9.1.7 [36] with a maximum number of iterations of 8.

### 2.2. Evaluation of Secondary Structure, Conserved Domains, and Post-Translational Modifications

The secondary structure of LmJean3 was predicted using the EMBOSS v6.5.7 tool [37] implemented in Geneious v9.1.7. Conserved domain identification was performed using the Conserved Domains Database (CDD) [38,39]. LmJean3 disordered regions were inspected by the Meta-Disorder method from the PredictProtein server [40]. The kinase active site (Prosite pattern PS00108: [LIVMFYC]-x-[HY]-x-D-[LIVMFY]-K-x(2)-N-[LIVMFYCT](3)) and ATP-binding site (Prosite pattern PS00107: [LIV]-G-{P}min-G-{P}-[FYWMGSTNH]-[SGA]-{PW}-[LIVCAT]-{PD}-x-[GSTACLIVMFY]-x(5,18)-[LIVMFYWCSTAR]-[AIVP]-[LIVMFAGCKR]-K) were determined by InterPro: Protein sequence analysis and classification [41]. Kinase active residues were predicted using the Pfam database from the European Bioinformatics Institute [42] (Uniprot accession number: Q4QBR6). EF-hand motifs were scanned using the PATTINPROT tool from the PRABI-Lyon-Gerland Network Protein Sequence analysis (NPS@) Web server (https://npsa-prabi.ibcp.fr). Prediction of EF-hand motifs was performed by searching for the motif signature of the canonical EF loop with both flanking helices described by Zhou et al., (2006) (x-{DNQ}-x(2)-{GP}-{ENPQS}-x(2)-{DPQR}-[DNS]-x-[DNS]-{FLIVWY}-[DNESTG]-[DNQGHRK]-{GP}-[LIVMC]-[DENQSTAGC]-x(2)-[ED]-[FLYMVIW]-x(2)-{NPS}-{DENQ}-x(3)) [43].

The following tools were used to assess post-translational modifications: The NetNGlyc 1.0 server from the Technical University of Denmark was used for N-terminal glycosylation prediction (http://www.cbs.dtu.dk/services/NetNGlyc/). The MYR Predictor of the IMP Bioinformatics Group assessed Jean3 sequences for possible myristoylation patterns (http://mendel.imp.ac.at/myristate/SUPLpredictor.htm). The TMHMM Server v2.0 from the Technical University of Denmark was used to determine protein transmembrane regions (http://www.cbs.dtu.dk/services/TMHMM/). PredGPI [44] was used to investigate the presence of GPI anchor sites. ProtParam [45] was used to predict the molecular mass of the protein and its theoretical isoelectric point. Serine and threonine phosphorylation sites were assessed by neural network prediction using the NetPhos 2.0 server [46]. Prediction of the presence and location of signal peptide cleavage sites was performed using the SignalP 4.1 server [47].

Genomic sequence distribution, multiple alignments, and plasmid diagrams were computed and arranged using Geneious version 9.1.7 created by Biomatters. Schematic representation of the protein with its motifs, regions, and conserved domains was drawn using Inkscape software v0.91 [48].

### 2.3. Phylogenetic Analysis

The phylogenetic tree was constructed with the sequences retrieved from the NCBI protein database and TriTrypDB [34]. LmJean3 (XP_001683232) full deduced amino acid sequence and individual conserved domains were used to perform BLASTp searches within the non-redundant protein sequences database (nr), with the default search parameters. Sequences were then aligned using the MAFFT algorithm [49] implemented in Geneious software, with a JTT200 scoring matrix.

ProtTest 2.4 server [50] was used to determine the best-fit model for the phylogenetic reconstruction of Jean3. According to ProtTest, the best model for protein evolution of the 23 sequences included in the analysis, with the smallest Akaike Information Criterion (AIC), was Le and Gascuel (LG) [51] with the following parameters: gamma shape (4 rate categories) of 1.295, proportion of invariable sites of 0.035; the observed amino acid frequencies obtained from the alignment were: A: 0.066, C: 0.016, D: 0.053, E: 0.069, F: 0.037, G: 0.066, H: 0.031, I: 0.060, K: 0.059, L: 0.100, M: 0.023, N: 0.045, P: 0.048, Q: 0.030, R: 0.061, S: 0.083, T: 0.047, V: 0.068, W: 0.007, and Y: 0.030. Geneious was used to portray the Newick output from the best tree (with optimized topology and branch lengths).

### 2.4. Screening for Homology Modeling Templates

The 262-amino acid sequence corresponding to the Ser/Thr domain of LmJean3 was used to perform a BLAST search within the Swiss-Model Repository [52]. Sequence alignment of the kinase domains from CIPK23, CIPK24/SOS2, Snf1, LdBPK_220630.1.1 (LdJean3), LbrM.22.0750 (LbJean3), and LmjF.22.0810 (LmJean3) was performed using the ClustalW iterative algorithm [53].

### 2.5. Molecular Modeling and 3D-Structure Validation

For homology modeling, the prediction server Robetta [54] was employed. Robetta uses the ROSETTA software to predict structures under the Ginzu protocol [54] either by homology modeling or by ab initio modeling. The C-terminal 97 amino acids (disordered region) from LmJean3 sequence were removed to improve the template selection by the software and consequently the final model. Ginzu alignment from the resultant N-terminal 275 amino acids from LmJean3 showed that the best candidate (0.8112 confidence) for comparative modeling was the B-chain from the 4CZT entry from the Protein Data Bank (PDB) [55]. 4CZT corresponds to the crystal structure of the kinase domain of CIPK23, determined by Chaves-Sanjuan et al. (2014) using X-ray diffraction with 2.3 Å resolution [56].

The backbone C_α_ root-mean-square deviation (RMSD) between 4CZT and the LmJean3 structures obtained from Robetta was determined using PyMOL [57]. The model with the lowest RMSD value was chosen for energy minimization using the YASARA force field [58]. The final model solvation accessibility was assessed by SolvX (http://ekhidna.biocenter.helsinki.fi/solvx/start). The three-dimensional profile of LmJean3 was determined by Verify3D [59]. The absolute quality of the model was estimated with QMEAN6 [60,61,62,63], and the atomic empirical mean force potential was determined by atomic non-local environment assessment (ANOLEA) [64,65,66]. The Swiss-Model Protein Structure and Assessment tools server [52] was used to assess the stereochemical quality of the model. The correctness of residue prediction was measured using PROCHECK [67] with a minimum resolution of 2.5 Å; the empirical force field energy was evaluated by GROMOS [68]. DSSP was used to compare the predicted model with the structural motifs from the secondary structure [69,70]. The three-dimensional model and the template were also checked using PROSA [71,72]. Finally, the structure quality of the template and the predicted LmJean3 structure was estimated using ERRAT [73]. All ribbon figures were designed using PyMOL v1.8.4.0 (https://sourceforge.net/projects/pymol/files/pymol/1.8/).

### 2.6. MD Simulation

The OPLS-AA force field [74] for the protein and the SPC water model [75] were employed. The predicted structure of LmJean3 was solvated in a rhombic dodecahedron box of water. The distance between the structure and the edge of the box was set to 12 Å, resulting in a minimal distance between periodic images of at least 24 Å. The net charge of the protein was neutralized by adding three positive counter-ions. Periodic boundaries were implemented in all three dimensions. Energy minimization of 1373 steps using the steepest descent algorithm was followed by two equilibration phases for 200 ps (first NVT—constant system, volume and temperature and then NPT—constant system, pressure and temperature ensemble). The temperature and pressure were controlled by a V-rescale thermostat and a Parrinello-Rahman barostat, respectively [76,77]. The equilibration phase was followed by 20 ns MD simulations, carried out using GROMACS v5.1.1 [78]. The integration time-step was of 2 fs, and the atom positions of the solute were written to file every 10 ps (every 5000 steps). To check the stability of the simulations, the RMSDs of the C_α_ atoms with respect to the minimized starting structure were calculated and monitored.

Data evaluation from MD simulations was carried out using the software package VMD v1.9.3 [79]. Salt bridges were identified by the following criteria: distance between any of the oxygen atoms of acidic residues and the nitrogen atoms of basic residues found within 3.2 Å, and centroids of the sidechain charged groups found within 4.0 Å of each other. Then, the trajectories were analyzed, and the presence of the identified salt bridges was assessed in each frame of the simulation. Similarly, hydrogen bonds were calculated with a cutoff distance of 3.2 Å and a cutoff angle of 20 degrees. A list was generated with those hydrogen bonds found in at least 30% of the trajectory (Appendix A). Residue interactions previously identified as salt bridges were not considered for hydrogen bond evaluation. In addition, the average coordinates over the trajectory were calculated and used to write a PDB file using VMD. Then, the RMSDs of the structure of the initial frame and the average structure were calculated using PyMOL v1.8.4.0.

### 2.7. Protein Preparation, Binding Site Identification, and High-Throughput in Silico Docking

Binding site identification, protein preparation, ligand preparation, and Glide docking were performed using Schrödinger Suite (Release 2016-1). Using Schrödinger’s protein preparation wizard [80], hydrogen atoms were added after deleting any original ones, followed by adjustment of bond orders for amino acid residues and ligand. The protonation and tautomeric states of Asp, Glu, Arg, Lys, and His were adjusted to match the physiological pH of 7.4. Possible orientations of Asn and Gln residues were generated. Hydrogen bond optimization (ProtAssign) sampling was performed using PROPKA [81] at a pH of 7.4.

Binding sites were identified using Schrödinger SiteMap v3.8 [82,83]. The region corresponding to the peptide substrate binding site was selected for grid files generation with Glide v7.0 [84,85]. The generated grid was used for in silico docking of small molecular compounds from the Zim dataset from ZINC [86] using Schrödinger’s Virtual Screening Workflow (VSW) with Epik penalties turned on [87,88]. From the results obtained, the molecular structures with the highest predicted affinity and lowest docking scores to the active site were selected (within Glide’s reported error of 2 kcal/mol).

### 2.8. Parasite Culture Conditions

*L. major* (Lv39c5) promastigotes were grown at 26 °C in M199 medium (Sigma-Aldrich, St. Louis, MO, USA) supplemented with 25 mM HEPES (pH 7.2; Sigma-Aldrich), 0.1 mM adenine (Sigma-Aldrich), 0.0005% (*w*/*v*) hemin (Sigma-Aldrich), 2 mg/mL biopterin (Sigma-Aldrich), 0.0001% (*w*/*v*) biotin (Sigma-Aldrich), 10% (*v*/*v*) heat-inactivated fetal bovine serum (Gibco Laboratories, Grand Island, NY, USA), and an antibiotic cocktail (50 U/mL penicillin, 50 mg/mL streptomycin) (Sigma-Aldrich). *L. major* cultures used for qPCR analysis were grown in Schneider’s medium (Gibco Laboratories) supplemented with 10% (*v*/*v*) heat-inactivated fetal bovine serum (Gibco Laboratories) and 40 μg/mL gentamicin (Sigma) at 26 °C.

### 2.9. Genetic Manipulation of L. major

The plasmid pXG-mCherry12 was constructed by our group as previously reported [89]. *L. major* DNA was extracted following the protocol previously described by Medina-Acosta et al., (1993) [90]. Then, the coding DNA sequence (CDS) corresponding to LmJean3 was amplified by PCR from *L. major* genomic DNA using the primers J3XStop-Fw and J3XStop-Rv for the construct pXG-LmJean3-mCherry12 (Appendix A), J3NotI-Fw and J3NotI-Rv for pXG-GFP^2+^-LmJean3 (Appendix A), and J3SF and J3SR for pXG-Hyg-LmJean3 (Appendix A). Primer sequences are detailed in Appendix A. Each of LmJean3 PCR products was then ligated into pCR^®^2.1-TOPO^®^ (ThermoFisher Scientific, Rockville, MD, USA) cloning vectors following the manufacturer’s protocol. The ligation products were used to transform DH5α *Escherichia coli* bacteria by the heat shock method. Plasmids were then extracted from the bacteria and digested with the following restriction enzymes: BstXI (Clonetech, Palo Alto, CA, USA) for the pXG-LmJean3-mCherry12 construct (Appendix A), NotI-HF (New England Biolabs, Massachusetts, USA) for the pXG-GFP^2+^-LmJean3 construct (Appendix A), or SmaI (Clonetech) for the pXG-Hyg-LmJean3 plasmid (Appendix A). Digestions were then gel-purified, and LmJean3 sequences were ligated into the cloning cage of pXG-mCherry12, pXG-GFP^2+^ or pXG-Hyg plasmids (Appendix A). LmJean3 sequence and orientation within the constructed vectors, pXG-LmJean3-mCherry12, pXG-GFP^2+^-LmJean3, and pXG-Hyg-LmJean3, were assessed by DNA sequencing and PCR, respectively.

A total number of 10^8^ log-phase *L. major* parasites were used to transfect the plasmids pXG-LmJean3-mCherry12, pXG-GFP^2+^-LmJean3, and pXG-Hyg-LmJean3 by electroporation. A BioRad Gene Pulser II machine (Bio-Rad Laboratories, Hercules, CA, USA) was used for electroporation, following a method previously described by Cruz et al., (1991) [91]. Recombinant colonies were isolated from M199 agar plates supplemented with 100 μg/mL of hygromycin B Gold (InvivoGen Europe, Toulouse, France) for pXG-LmJean3-mCherry12 and pXG-Hyg-LmJean3, or 30 μg/mL geneticin (Sigma Aldrich) for pXG-GFP^2+^-LmJean3.

### 2.10. Fluorescence Microscopy

A final number of 2 × 10^7^ log-phase pXG-LmJean3-mCherry12 and pXG-GFP^2+^-LmJean3 *L. major* promastigotes were harvested, centrifuged at 7000× *g* for 10 min, and then fixed using a 1% paraformaldehyde/PBS solution. The pellets were then washed twice with PBS, and the cells were resuspended in a 1 mg/mL DAPI (Sigma Aldrich, USA) solution for 30 min at 4 °C for kinetoplast and nucleus staining. The parasites were washed twice with PBS before visualization. Images from the slides were acquired using a PerkinElmer ultraVIEW confocal microscope, employing 405/400–600 nm, 488/500–560 nm, and 561/570–700 nm excitation/emission wavelengths.

### 2.11. Metacyclic Forms Isolation

Metacyclic parasites (unagglutinated cells) were isolated using the peanut agglutinin (PNA) method (Sacks et al., 1985) from 5 × 10^8^ stationary parasites [92]. The cells were then counted using a Z1 Coulter counter (Beckman Coulter, Fullerton, CA, USA) and used for the extraction of their RNA.

### 2.12. In Vitro Infections

A total number of 3 × 10^5^ RAW 264.7 murine macrophages per well were seeded in 6-well plates and grown at 37 °C in DMEM medium (Gibco Laboratories) supplemented with 10% (*v*/*v*) heat-inactivated FBS (Gibco Laboratories) and an antibiotic cocktail (50 U/mL penicillin, 50 mg/mL streptomycin) (Sigma). Macrophages were infected by pXG-LmJean3 and pXG-Hyg metacyclic parasites isolated by the PNA method [92] using a 25:1 (parasites/macrophage) ratio. The plates were incubated at 37 °C in a 5% CO_2_ atmosphere. After 24 h, non-phagocytosed parasites were washed with PBS, and fresh medium was added to the wells. The cells were harvested after 48 h by incubation in a trypsin–EDTA solution (Gibco Laboratories) for 5 min at 37 °C. Finally, the cells were centrifuged at 300× *g* for 10 min, and the pellets were washed twice with PBS before RNA extraction.

### 2.13. RNA Expression Quantification

For wild-type (WT) gene quantification, the RNA from log phase and stationary phase procyclic and metacyclic parasites from the in vitro infections of RAW 264.7 macrophages was extracted using a Qiagen RNAeasy extraction mini kit (Qiagen, Hilden, Germany). For the quantification of gene expression from transgenic parasites, the RNA from log-phase procyclic parasites was also extracted with the same kit.

RNA samples were then treated with Ambion DNA-free Kit (Invitrogen, Vilnius, Lithuania) following the manufacturer’s instructions. Retrotranscription was performed using 800 ng of RNA and M-MLV Reverse Transcriptase (Promega, Madison, WI, USA), following the protocol of the manufacturer. The obtained cDNA was then used for the qPCR assay. qPCR assays were performed using a 7500 real-time PCR system (Applied Biosystems, Foster City, CA, USA), 96-well plates (Applied Biosystems), and SYBR Green PCR master mix (Applied Biosystems). The primers used for qPCR are listed in Appendix A. The expression levels of glyceraldehyde-3-phosphate dehydrogenase (GAPDH) were used to normalize gene expression.

### 2.14. Cell Cycle Analysis by Propidium Iodide Staining and Flow Cytometry

A total amount of 10^6^ log-phase parasites were harvested, washed twice with cold PBS, and fixed with 70% ethanol overnight. Then, the cells were centrifuged and resuspended in a solution of PBS with RNase A (Sigma) at a final concentration of 50 µg/mL for 20 min (37 °C). The cell suspensions were centrifuged and washed twice with PBS; the parasites were then stained with a 5 µg/mL solution of propidium iodide (Sigma). Finally, the cell suspensions were subjected to flow cytometry using an Attune flow cytometer (Applied Biosystems, CA, USA). Ten thousand events were collected for each sample. The number of gated cells in the G1, G2/M, and S-phase is presented as a percentage.

### 2.15. Cytotoxicity Evaluation

Log-phase *L. major* parasites were seeded in 96-well plates and grown at 26 °C with increasing concentrations of amphotericin B (Sigma), miltefosine (Calbiochem, Darmstadt, Germany), paromomycin (Sigma), and geneticin (Sigma), diluted in M199 medium. After 48 and 72 h of incubation, parasites’ viability was measured using 3-[4,5-dimethylthiazol-2-yl]-2,5-diphenyl-tetrazolium bromide (MTT), as previously described [93]. The half-maximal effective concentration (EC_50_) was determined by fitting a sigmoidal Emax model to dose–response curves. Promastigotes viability was evaluated by comparison with untreated control cells. The results were expressed as means ± standard deviation (SD).

### 2.16. Statistical Analysis

Statistical analyses were carried out with GraphPad Prism v7.0b [94]. Two-group comparisons were performed by employing unpaired, two-tailed Student’s *t*-tests; *p* values < 0.05 were considered statistically significant (* *p* < 0.05, ** *p* < 0.01, *** *p* < 0.001).

## 3. Results

### 3.1. LmjF.22.0810 (LmJean3), a Novel Predicted Trypanosomatid Protein Kinase

LmjF.22.0810 was identified and retrieved from GeneDB (www.genedb.org) as one of the *L. major* genes encoding proteins harboring a kinase catalytic domain of the CAMK family (determined by phylogenetic inference [22]). Interestingly and on the basis of previous studies, LmjF.22.0810 had been described as a possible drug target [28]. LmjF.22.0810 orthologs are located on chromosome 22 of all *Leishmania* species whose genome was sequenced and included in TriTrypDB (www.tritrypdb.org). Gene sequence alignments displayed a high identity among *Leishmania* spp. (>85%) (Appendix A). This novel gene encoding a protein with a kinase domain was presently named ‘LmJean3′. Finally, Sanger sequencing of LmJean3 from *L. major* (Lv39c5) was performed and showed a single nucleotide mutation (C282T) when compared to the gene sequence from the database (Appendix A). Nevertheless, this polymorphism (C282T) does not change the translated protein sequence, given that both codons (AUC and AUT) encode the same amino acid (isoleucine).

Genes encoding protein orthologs were also found in *T. cruzi* and *T. brucei* on chromosomes 13 and 7, respectively. Gene sequence alignments displayed more than 50% identity when compared to the *Trypanosoma* spp. genes (Appendix A).

### 3.2. LmJean3 (LmjF.22.0810) Domains and Motifs

LmJean3 is a 372-amino acid protein with an estimated molecular mass of 40.87 kDa and a theoretical pI of 6.60. A putative protein kinase domain was found between residues 8 and 266 (Figure 1A), containing all the currently known essential motifs and residues required for protein kinase activity among its 11 kinase subdomains (Appendix A). Moreover, we also identified several sites and regions involved in the phosphorylation activity of kinases, such as ATP-binding site, catalytic loop, and activation segment (Figure 1A). No putative glycosylation, myristoylation, signal peptide, or GPI anchor was found within the sequence of LmJean3. Nonetheless, sequence analysis revealed EF-hand patterns at two C-terminal regions of the protein (244–272 and 333–361, with a percentage of similarity to the query PROSITE pattern of 83% and 81%, respectively; Figure 1A).

Additionally, we identified 17 serine and three threonine residues predicted to be putative phosphorylation sites in LmJean3 (Appendix A). Two of those residues were found within the activation loop (S157 and S162). Furthermore, the region containing these residues (157S–X–X–X–X–S162) is conserved within *Leishmania* and shows similarity to the motifs of MAPKKs of animal, yeast (S–X–X–X–S/T), and plants (S/T–X–X–X–X–X–S/T) that turn the protein into the active state after phosphorylation [95]. On the other hand, in *T. cruzi* and *T. brucei,* the C-terminal serine from the motif is different. In *T. cruzi,* a glutamine is found at this position (Q161), while in *T. brucei*, a glutamate is present (E161), which may mimic the required phosphoresidue, analogously to other kinases such as Phk [96].

The kinase domain is followed by a less conserved disordered region (residues 275–372) (Figure 1A) that did not display significant similarity to other proteins after BLAST analysis. Furthermore, most of the amino acid substitutions may occur at the C-terminal region of Jean3 sequences, while the N-terminal region where the kinase domain is located is highly conserved (Figure 1B). On the contrary, *Leishmania braziliensis* Jean3 protein displayed at its N-terminal region an additional extension of 60 amino acids compared to its homologs from other trypanosomatids (Figure 1B).

### 3.3. The Phylogeny of LmjF.22.0810 Homologues

The kinase domains of non-kinetoplastid organisms (*Arabidopsis thaliana, Homo sapiens,* and *Saccharomyces cerevisiae*) were retrieved from a BLASTp search and were used to reconstruct the phylogram of LmjF.22.0810 homologues. Such phylogenetic reconstruction not only supported the aforementioned homology among trypanosomatids but also confirmed that LmjF.22.0810 orthologs are restricted to kinetoplastid organisms (Figure 1C). The Ser/Thr kinase domains of CIPK23 (calcineurin B-like interacting protein kinase 23), SOS2 (salt overly sensitive 2), Snfp1 (sucrose non-fermenting kinase 1), AMPK (AMP-activated protein kinase), MARK2 (microtubule affinity-regulating kinase 2), and TSSK3 (testis-specific serine kinase 3) displayed sequence homology to the catalytic domain of LmJean3 as determined by BLAST analysis and multiple-sequence alignment. However, the phylogenetic reconstruction did not show a clear orthologous relationship with other specific ePKs, except for those in the kinetoplastids branch (Figure 1C). The phylogenetic reconstruction also included the protein CDPK1 (calcium-dependent protein kinase 1) from *Plasmodium falciparum,* an extensively characterized protein with EF-hand motifs within its sequence [98]. As shown in Figure 1C and previously observed [22], LmJean3 kinase and its homologues do not belong to the clustering of CDPK.

### 3.4. LmJean3 was Localized in the Cytoplasm, Nucleus, and Flagellum of Leishmania Promastigotes and Was Significantly Expressed in the Infective and Amastigote Forms

The red and green fluorescent fusion protein constructs pXG-LmJean3-mCherry12 and pXG-GFP^2+^-*LmJean3* were used to locate LmJean3 inside transgenic *L. major* parasites. Both fluorescent approaches (LmJean3-mCherry12 and GFP^2+^-LmJean3) allowed the visualization of LmJean3 in the cytosol, nucleus, and flagellum of the parasites (Figure 2A). Despite the observed localization of this *L. major* protein, analysis of the previously mentioned additional N-terminal 60-residue fragment from *L. braziliensis* by the TMHMM Server predicted a transmembrane region exclusive to the *L. braziliensis* protein (Appendix A).

The expression levels of LmJean3 were quantified during the in vitro growth of *L. major* promastigotes and also from amastigotes derived from in vitro infections. The gene expression levels of LmJean3 increased ~2-fold for log-phase parasites as they reached the stationary phase as procyclic or metacyclic promastigotes (Figure 2B). After that, these expression levels were maintained in the infective form (metacyclic promastigotes) and the amastigote parasites (Figure 2B).

### 3.5. The Kinase Domain of LmJean3 Is Conserved and Exhibits a High Similarity to CIPK24/SOS2, CIPK23, and Snf1 Kinase Domains

Since no crystallographic structure of LmJean3 kinase homologues was available, homology modeling was chosen as an approach to generate the protein structure of LmJean3. The most suitable templates found in the Swiss-Model Repository were two CIPK protein kinases from *A. thaliana* (CIPK23 and CIPK24/SOS2) and Snf1 from *S. cerevisiae*. SOS2 and Snf1 are two kinases that belong to the CAMK family. The multiple-sequence alignment of critical regions for kinase phosphotransferase activity is shown in Figure 3. The catalytic loop, activation loop, glycine-rich loop, and ATP-binding region signature of Jean3 kinases were found to share a high homology with the possible templates (Figure 3). Moreover, catalytically important residues for kinases, such as R128 and D129 from the H–X–D triad [99], as well as D147 from the DFG (Asp–Phe–Gly) triplet, K37 from the A–X–K motif, and E53 from the C-helix [100], were conserved in all the analyzed sequences (Figure 3). Sequence identity was 34.22% between LmJean3 and Snf1, 37.26% between LmJean3 and CIPK23, and 41.83% between LmJean3 and SOS2. Consequently, sequence analysis indicated the protein structures of CIPK23, CIPK24/SOS2, and Snf1 represented suitable templates for homology modeling. The complete sequence alignment can be found in the Appendix A.

### 3.6. Homology Modeling, Refinement, and Validation of LmJean3 Structure

The Robetta webserver was used to generate the tertiary structure of the protein [54]. After Ginzu alignment of the LmJean3 kinase domain, Robetta determined CIPK23 as the best template for homology modeling (0.81 confidence). Consequently, an intermediate model was generated based on chain B of the 4CZT entry in the Protein Data Bank (PDB) repository. 4CZT corresponds to the crystal structure of the kinase domain of CIPK23 determined by Chaves-Sanjuan et al. (2014) using X-ray diffraction with a resolution of 2.3 Å [56]. Finally, energy minimization was performed with the YASARA force field to alleviate steric clashes and other peculiarities in the intermediate model [58]. The optimized resulting model was then validated. ERRAT determined the overall quality factor of the final model to be 98.11% (Figure 4A), as opposed to 95.49% of the template (4CZT) (Figure 4B). ProSA displayed a similar overall quality (Z-score) for LmJean3 (−7.66; Figure 4C) and the template (−8.47; Figure 4D). Negative Z-scores are required to consider the validity of structures [102]. The Ramachandran plot showed 88.4% of the residues to be localized in the favored regions, while none was found in the disallowed regions (Figure 4E).

Local model quality estimations by GROMOS and ANOLEA, in addition to LmJean3 structural features determined by DSSP [64,68,69,70], were also evaluated (Appendix A). These data confirmed the validity of the model. Finally, the observed backbone C_α_ RMSD between LmJean3 and the template (4CZT_B) was 1.87 Å, while superposition of the five active site relevant residues (K37, E53, R128, D129, and D147) displayed an RMSD value of 0.58 Å (Figure 5A). Thus, the catalytically relevant residues were aligned to their equivalents in the template. A summary of the global quality of the optimized predicted structure is shown in Appendix A.

### 3.7. Overall Description of the Predicted Structure of LmJean3 Catalytic Domain

The structure of the predicted catalytic domain of LmJean3 displayed a canonical Ser/Thr kinase fold and was in accordance with the template used for homology modeling. The protein folded into two lobes (Figure 5B), with the catalytic cleft located between them. The core of the N-lobe consisted of five-stranded antiparallel β-sheets (β1-β5) and the C-helix. The glycine-rich loop (residues 15–20) was found within the N-lobe between β-strands 1b and 2. On the other hand, the C-lobe comprised six well-defined helices (αD-αI), with the catalytic loop (residues 125–137) and the activation loop (residues 147–180) within them. Inside the C-lobe, β8 and αT1 were found flanking the DFG motif, where D147 was inferred to recognize one of the ATP-bound Mg^2+^ ions [103,104]. Residues R128 and D129 from the H–X–D triplet were found inside the catalytic cleft, within the peptide substrate binding region (Figure 5B). Finally, the activation loop, which is delimited by two α-helical turns (αT1 and αT2), was found placing the G-loop aside from the active site (Figure 5B). This disposition of the activation loop impedes the access of protein substrates to the catalytic cleft and is a major indicator of the inactive state of a kinase [104]. Since kinase conformations are determinant for the specificity of small-molecule inhibitors [14,30,31,105], we decided to assess the local spatial pattern (LSP) alignment of LmJean3 catalytic and regulatory spines. LSP showed the alignment between both spines to be broken and the C-helix directed outward from the active site (αC-out) (Figure 5C). Additionally, as shown in Figure 5A, D147 from the DFG triplet appeared directed towards the ATP-binding site in what is known as the “DFG-in” conformation. Therefore, the LmJean3 structure displays a DFG-in/αC-out inactive conformational state, in agreement with the conformation of the template.

### 3.8. Molecular Dynamics Simulation

Over the course of the simulation, the RMSD per residue was calculated against the initial frame. The protein RMSD increased to a maximum deviation of 3.89 Å after 15.34 ns, with an average value of 2.94 ± 0.52 Å (Appendix A). Importantly, the observed RMSD for the catalytically relevant residues of LmJean3 (K37, E53, R128, D129, and D147) was rather constant during the simulation (Appendix A), having a maximum of 1.47 Å at 14.67 ns and an average value of 0.89 ± 0.15 Å.

The structure obtained from the average atomic coordinates of the trajectory was aligned against the predicted catalytic residues of the initial LmJean3 structure (Figure 5D). The catalytically relevant residues of both structures displayed an RMSD value of 1.15 Å. Hence, their structural resemblance throughout the duration of the simulation provides confidence in the homology model structure.

Non-covalent residue interactions such as salt bridges and hydrogen bonds were also calculated during the simulation (Appendix A). The main interactions found within the kinase domain and those involving the catalytically relevant residues, as well as their persistence during the trajectory, are described in Table 1. The observed non-covalent interactions for LmJean3 were similar to those reported for other kinases such as Src, PKA, and Twitching [106,107,108]. Figure 5E shows that R254 from subdomain XI formed a highly conserved stabilizing salt bridge with E180 from the APE motif [107]. Similarly, an intracatalytic salt bridge, known for stabilizing the catalytic loop, was observed in subdomain VIb between K131 and D129 (Table 1; Figure 5E) [107,108]. As a consequence of the αC-out conformation found in the structure, R152 from the activation segment and E53 from the αC-helix were found bound by a salt bridge [108]. Conversely, K37 from the A–X–K motif formed a salt bridge with D147, indicating an inward direction of DFG–Asp towards the active site, a feature known as one of the prerequisites for switching into the active state (“DFG-in” conformation) (Table 1; Figure 5E). Additionally, residues L137–G88, R128–D192, and N134–D129 displayed several stabilizing hydrogen bonds with a persistence higher than 40% (Table 1), equivalent to interactions reported for other protein kinases [106,107,108,109].

### 3.9. LmJean3 Binding Sites: Prediction and Analysis

The top-five ranked potential ligand-binding sites of LmJean3 (named site A, B, C, D, and E) were identified using SiteMap v3.8 [82,83] and are summarized in Appendix A. Site A (SiteScore = 1.05) corresponds to the solvent-exposed regions of the allosteric site, the ATP-binding pocket, and deeper cavities of the ATP site (Figure 6A). Site B (SiteScore = 0.92) is located in a smaller allosteric cleft found at the protein substrate binding site (Figure 6B). This site includes C-lobe residues of the catalytic loop, the activation loop, the P+1 loop, A188, A191, and S195 of the F-helix, and D49 and V46 of the αC-helix (Figure 6B). Interestingly, residues of the catalytic loop were found in Site B, not obstructed by the activation loop. The distribution of Site A appeared wider than that of Site B, similarly to what has been described for other DFG-in/αC-out conformations [30]. Both sites (A and B) displayed more hydrophilic than hydrophobic components (Figure 6 and Appendix A). On the other hand, sites C, D, and E did not surpass the recommended SiteScore cut-off of 0.80 [83]. The druggability (Dscore) of sites A and B was also evaluated using SiteMap. Site A displayed a Dscore of 1.03, and Site B a Dscore of 0.85, both being promising values for targeting by small-molecule, drug-like ligands [83]. Due to the low selectivity that ATP-binding-site inhibitors typically display [14,30,31,104] and to minimize any off-target activity [18], we decided to target the allosteric site found in the catalytic pocket of LmJean3 (Figure 6B).

### 3.10. Aminoglycosides Are Predicted Ligands of LmJean3.

More than 11,400 molecules from the Zim dataset (April 2016) from ZINC [86] were used for in silico docking to the Site B of LmJean3. Zim (ZINC in man) is a subset of experimental compounds, including drugs, which have been approved for use in humans. Eleven compounds were predicted to bind LmJean3, with predicted docking scores ranging from −9.06 to −11.46 kcal/mol (within GLIDE’s error margin of 2 kcal/mol) (Appendix A). Seven of the best solutions were AG antibiotics, three were flavonoids, and one was the anthracenedione mitoxantrone. The seven best docking solutions were ranked by their calculated docking score (Table 2). Limitations of the database, such as stereochemical ambiguity [86], were considered. In fact, only stereoisomers of neomycin B (NEO) and amikacin (AMK) were predicted to bind to Site B (Table 2). On the other hand, paromomycin (PMM) and tobramycin (TBR) were docked as their natural purified structures (Table 2).

Four of the docked compounds from Table 2 are either AGs (PMM and TBR) or isomers of AGs (NEO and AMK). In *Leishmania*, AGs are supposed to bind directly to the eukaryotic A-site of the small ribosomal subunit (40S) [110]; however, their mechanism of action is not completely clear. The four docked AGs contain a 2-deoxystreptamine core (2-DOS; Ring II; Figure 7) and can be classified on the basis of their substitution positions in the 2-DOS ring. PMM and NEO 2-DOS groups are substituted at positions 4 and 5 (Figure 7A), whereas AMK and TBR have a 4,6-disubstituted 2-DOS group (Figure 7B). Additionally, only PMM displays a 6′-OH substituent group at Ring I, while the other docked AGs are 6′-NH_2_ derivatives (green-colored circles in Figure 7). The nature of the substituent group at the 6′-position of Ring I has been described to be an important component of selectivity toward eukaryotic species [111]. The aforementioned statement is in agreement with previous works in *Leishmania* according to which NEO, AMK, and TBR (6′-NH_2_ derivatives) displayed either high EC_50_ values in vitro or incomplete healing in vivo, in contrast to PMM and geneticin (G418) which are 6′-OH derivatives [110,112,113].

### 3.11. Predicted Interactions of Paromomycin with Site B of LmJean3

The predicted interactions of PMM within Site B are presented in Figure 7C (DockScore of −11.46 kcal/mol). The 2-DOS ring from PMM was found hydrogen-bonded to residues from the H–X–D triplet, as well as to residues in the activation loop and the αC-helix. Three hydrogen bonds were found between R128 and the PMM 2-DOS ring (Ring II). Also, D129 formed a hydrogen bond and a salt bridge with the protonated amine moiety at position 1 from Ring II. R155 in the activation loop was found to form a hydrogen bond with the 6′-OH substituent group from Ring I, while PMM Ring IV was hydrogen-bonded to E156 and R152. Finally, D49 from the αC-helix displayed an H-bond and a salt bridge with position 6′′′ of Ring IV. Overall, these results showed that residues from the H–X–D triplet and C-terminal to the DFG motif, two key regions for substrate binding and γ-phosphate transfer [99,114,115], may play a crucial role in PMM binding to Site B.

### 3.12. Docking in Site A and Cross-Docking

The top-ranked molecules resulting from the Site B docking analysis as well as geneticin were also docked into Site A. Moreover, the detergent molecule CHAPS (C_32_H_58_N_2_O_7_S, HET-ID CPS, 3-[(3-cholamidopropyl)dimethylammonio]-1-propanesulfonate), found in the ATP-binding site of the template CIPK23 during the crystallization process, and ATP were also analyzed to predict their docking score to Site A (self-dock for ATP) and to Site B.

As expected, ATP showed a stronger docking score for Site A than for Site B (Figure 8). Geneticin displayed a weaker docking score for both sites as compared to those of other dataset molecules. On the other hand, PMM displayed the strongest docking score (lowest free energy) for both sites (Figure 8). These data show a predicted strong putative interaction between PMM and LmJean3. Such predicted interaction was stronger than that of other cofactors and possible ligands.

### 3.13. Generation of LmJean3-Overexpressing Parasites (LmJ3OE)

We next assessed the activity of some of the docked drugs onto *L. major* highly expressing LmJean3. By using the constructed expression vector pXG-LmJean3, we developed a transgenic L. major strain overexpressing LmJean3 (LmJ3OE). qPCR revealed that LmJ3OE parasites showed a 30-fold higher LmJean3 gene expression level compared to the controls (pXG-Hyg parasites) (Appendix A). In addition, the cell cycle distribution and growth rate of LmJ3OE parasites were similar to those observed for the controls (Appendix A).

The transcriptional profiles of several genes related to cell morphology, proliferation, and treatment resistance were also evaluated in promastigotes.

Expression levels of ABC-transporter genes (ABCA3, ABCH1, ATPase 1-like, ABCC2, and MRPA) and α-tubulin (an important component of cell cytoskeleton) were measured in log-phase promastigotes from LmJ3OE and control parasites (Appendix A). No significant alteration was detected between the two groups, with the exception of ABCA3 (LmjF.11.1240), which was significantly downregulated in LmJ3OE parasites (*p* < 0.001; Appendix A).

### 3.14. LmJ3OE Parasites Exhibited Less Sensitivity to Paromomycin and Other Aminoglycosides

Since PMM, a first-line treatment against leishmaniasis, was previously proposed to be a putative ligand of LmJean3, we investigated this hypothesis.

Through the analysis of the growth inhibition of LmJ3OE and control parasites after PMM treatment, EC_50_ values were estimated. Our data showed that LmJ3OE promastigotes were significantly less sensitive to PMM after 48 h and 72 h of exposure (~2-fold increase of EC_50_) (Figure 9A). In addition to PMM (a 4,5-disubstituted aminoglycoside), we also tested geneticin (a 4,6-disubstituted aminoglycoside and another 6′-OH derivative). Our results confirmed LmJ3OE parasites exhibited less sensitivity to paromomycin and geneticin (Figure 9A,B; Appendix A). On the contrary, compared to the controls, LmJ3OE promastigotes displayed a significantly higher sensitivity to both leishmanicidal compounds amphotericin B and miltefosine (Figure 9C,D; Appendix A), predicted not to interact with our target.

## 4. Discussion

PKs are key regulators of cellular signaling and have long been recognized as important drug targets in a variety of diseases (such as cancer and inflammatory and infectious diseases) [14,15,16,17]. LmjF.22.0810 was identified to encode a protein kinase that we named “LmJean3”. This PK harbors EF-hand sequences, which are intrinsic motifs for calcium-dependent protein kinases (CDPKs). CDPKs are a prominent group of kinases found in plants and parasites, such as *P. falciparum* and *Toxoplasma gondii,* but absent in human and yeast kinomes [22]. However, the phylogenetic inference of LmJean3 kinase homologues did not support its clustering with CDPKs. Additionally, the known regulatory motifs of CAMKs were also absent in LmJean3. Thus, as previously mentioned by Parsons et al. (2005)*,* it is likely that Jean3 genes encode a novel class of ePKs containing EF hands [22].

LmJean3 was found to be located in the flagellum, cytoplasm, and nucleus of *L. major*. On the basis of the high protein sequence identity found in all the *Leishmania* species analyzed, LmJean3 orthologs are expected to display a similar localization. Since trypanosomatids lack tyrosine kinases, it has been proposed that receptor Ser/Thr kinases could respond to host or parasite ligands and mediate intercellular communication [20,22]. Moreover, the predicted N-terminal transmembrane region of Jean3 from *L. braziliensis* may represent an additional feature that could be further evaluated in order to identify biological functions of LmJean3 kinase homologues. However, the primary focus for this research study was to characterize LmJean3 as a drug target.

In the first step, computations were performed to predict a 3D structure for LmJean3 suitable for molecular docking. The evaluation of the protein structure and molecular dynamics simulations showed that the LmJean3 model and its catalytic residues were stable, providing validation of the protein structure. We also showed in silico that PMM and TBR may bind to LmJean3 with multiple hydrogen bonds, which are major components of stable protein–ligand complexes [116]. Cross-docking indicated that the strength of the interaction between PMM and the protein is stronger than that of the interaction with other cofactors and possible ligands, including those co-crystallized in the original template. Finally, to complement the in silico studies, we evaluated the cytotoxicity of PMM and geneticin in transgenic LmJean3-overexpressing *L. major* parasites (LmJ3OE). Geneticin has an analog structure to TBR, with a 4,6-disubstituted 2-DOS ring; on the contrary, PMM bears a 4,5-disubstituted 2-DOS ring. Nonetheless, previous studies have shown that the leishmanicidal activity of both compounds is mainly due to their 6′-OH substituent group in Ring I [110]. In vitro tests determined that LmJ3OE parasites’ sensitivity is significantly lower to both drugs. In agreement with this finding, the overexpression of a specific target has been described as a putative mechanism for drug resistance in eukaryotic pathogens [6].

In bacteria, the enzymatic modification of AG antibiotics is the most prevalent type of resistance found in the clinical setting [3]. It may result from the addition of an acetyl, a nucleotydil, or a phosphate group by acetyltransferases, nucleotidyltransferases, or phosphotransferases, respectively [117]. It is remarkable that aminoglycoside phosphotransferases (APHs) are proteins that share structural and functional homology with ePKs but display very little primary sequence conservation (<5%) [2,118]. It is also interesting to highlight that some APHs from pathogenic bacteria confer resistance by sequestering AGs by tight binding, rather than by inactivation of the target [119,120]. On the basis of this strategy, and in agreement with the docking results and cytotoxicity evaluation, we postulate that Jean3 may be capable of sequestering PMM and geneticin.

Additionally, it has been proposed that PMM might not only interact with ribosomes but also alter membrane fluidity, interfere with the mitochondrial membrane potential, and inhibit respiration [121,122,123]. Hence, we measured the gene expression levels of PCNA, *α*-tubulin, and various ABC-transporters. In *Leishmania*, those molecules had previously been related to resistance against several drugs such as miltefosine, antimony, and PMM [122,124,125]. Interestingly, in the LmJ3OE strain, we found that *ABCA3* (LmjF.11.1240) was highly downregulated. *ABCA3* overexpression has been observed in antimony-resistant strains; however, transfection of the gene did not confer a drug-resistant phenotype, excluding it as the unique factor determining antimony resistance [126]. On the contrary to antimony-resistant strains, the expression of *ABCA3* has been observed to be inversely correlated with miltefosine susceptibility in *Leishmania* (*Viannia*) [127]. Nevertheless, in LmJ3OE parasites, the low expression of *ABCA3* did not correlate directly with their increased susceptibility to miltefosine. These data seem to indicate the existence of a broader mechanism underlying the observed drug response in LmJ3OE parasites.

Interestingly, the low number of cases of patients infected with PMM-resistant strains is linked to its occasional use as a clinical treatment for visceral leishmaniasis [7]. PMM has been used more widely for the treatment of the cutaneous disease, in which two cases caused by PMM-resistant *L. aethiopica* reported three- to five-fold drug sensitivity decrease [7,128]. A review of published research data from drug-resistant strains does not implicate *Jean3* transcript or its coding protein in PMM resistance. Therefore, it would be insightful to further analyze LmJean3 homologues transcripts in drug-resistant strains.

The predicted complexes of LmJean3 with different AGs within its protein binding site rationalize the structural basis for this enzyme broad-spectrum AG resistance activity. Furthermore, the generated predicted model of LmJean3 could be exploited for structure-based drug design of compounds to combat leishmaniasis. From the perspective of drug resistance, the examination of AG binding to LmJean3 homologues may highlight avenues for the development of next-generation aminoglycosidic drugs.

## Figures and Tables

**Figure 1 biomolecules-09-00723-f001:**
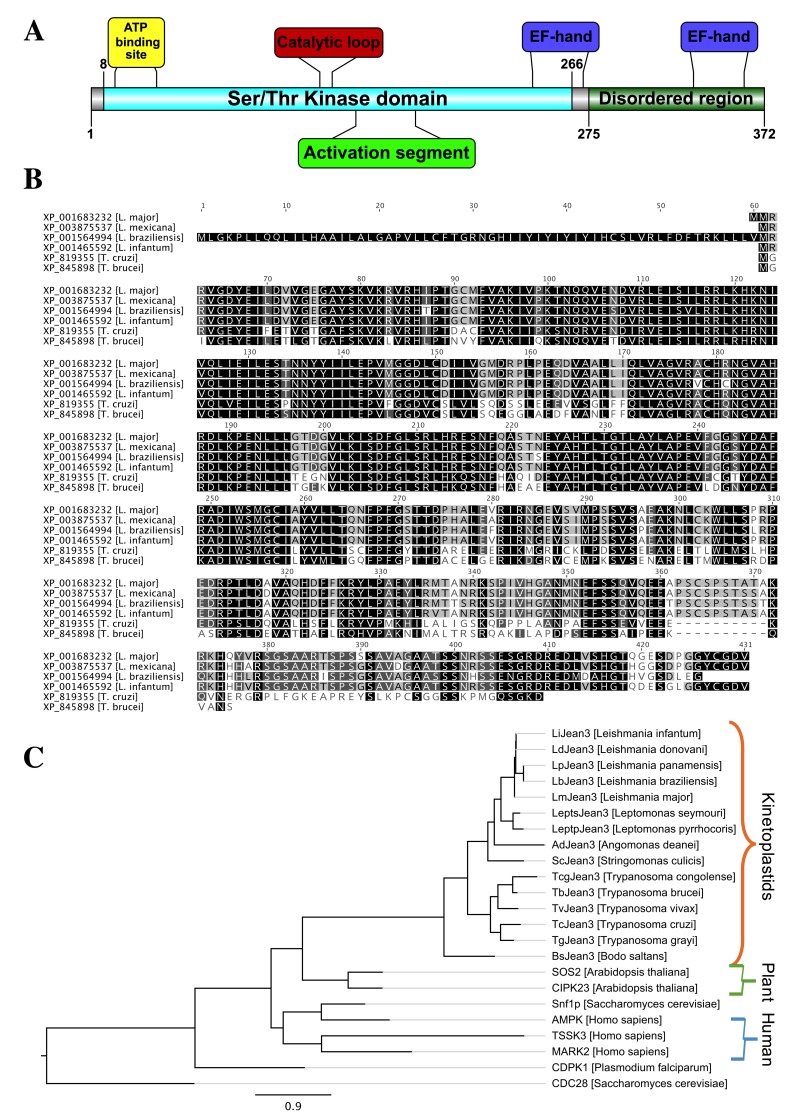
Homologues of LmjF.22.0810 (LmJean3) kinase are highly conserved in trypanosomatids. (**A**) Schematic representation of the predicted catalytically relevant regions and domains of LmJean3; (**B**) *Leishmania major*, *Leishmania mexicana*, *Leishmania braziliensis*, *Leishmania infantum*, *Trypanosoma cruzi*, and *Trypanosoma brucei* Jean3 orthologs sequences alignment. Residues are colored on the basis of their similarity under a BLOSUM62 score matrix [97]; (**C**) phylogenetic analysis of Jean3. *Saccharomyces cerevisiae* CDC28 was used to root the tree. Protein sequences from Jean3 trypanosomatid orthologues (accession numbers: XP_001465592, XP_003860813, XP_010699073, XP_001564994, XP_001683232, XP_015664142, KPI87780, EPY38836, CCC91397, XP_845898, CCD21423, XP_819355, XP_009309601, CUI14844, EPY19901) and kinases from *Arabidopsis thaliana* (CCH26589, NP_564353), *Homo sapiens* (AAA64745, NP_001156768, NP_443073), *S. cerevisiae* (NP_010765, NP_009718) and *Plasmodium falciparum* (P62343) were used to generate the phylogram.

**Figure 2 biomolecules-09-00723-f002:**
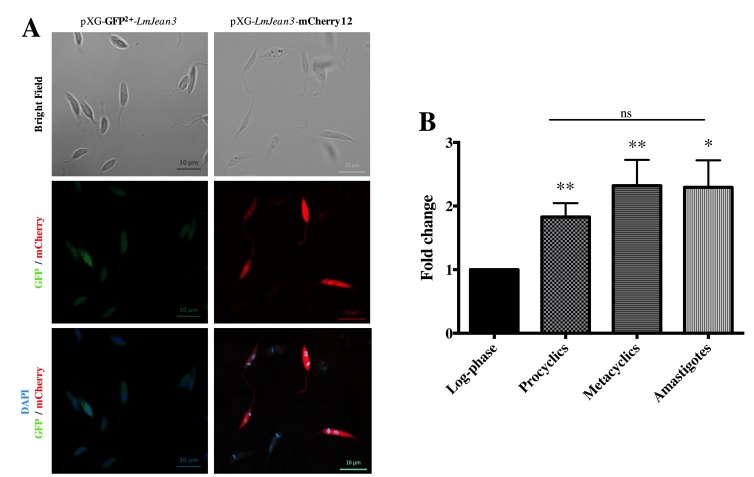
LmJean3 was localized in the cytoplasm of *L. major* parasites, and its gene expression level s doubled in the stationary phase. (**A**) Visualization of *L. major* promastigotes expressing GFP^2+^-LmJean3 (left lane) or LmJean3-mCherry (right lane) fusion proteins; (**B**) relative expression quantification of LmJean3 from log-phase, procyclic, and metacyclic (infective) promastigotes and from intracellular amastigotes. Bars represent gene expression mean fold change (± SD) from three independent experiments (* *p* < 0.05; ** *p* < 0.01).

**Figure 3 biomolecules-09-00723-f003:**
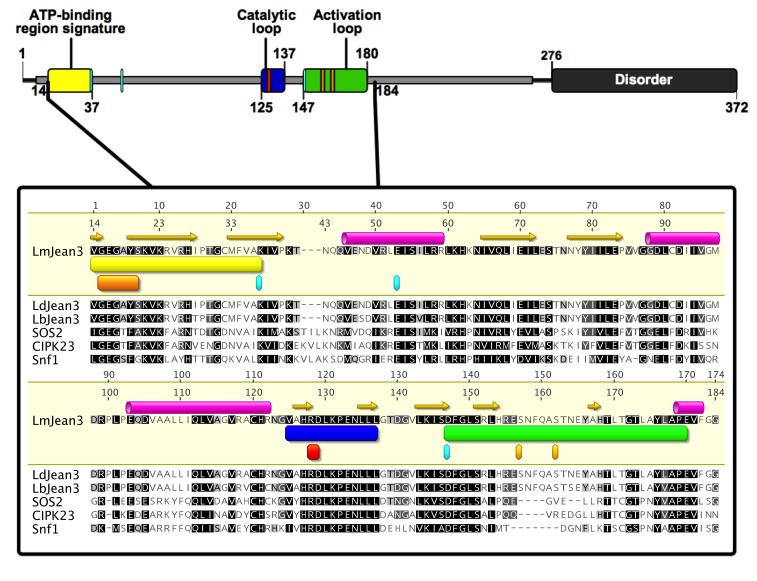
The kinase domain of LmJean3 homologues shares a high overall similarity to CIPK24/SOS2, CIPK23, and Snf1 kinase domains. On top: schematic representation of LmJean3 secondary structure, sequence, and regions. Underneath: LmJean3 sequence alignment (residues 88-211) with *Leishmania donovani* Jean3 (LdJean3), *L. braziliensis* Jean3 (LbJean3), *A. thaliana* CIPK24/SOS2, *A. thaliana* CIPK23, and *S. cerevisiae* Snf1. Within the ATP-binding region signature (yellow rectangle), the glycine-rich loop is highlighted as an orange rectangle (residues 15–20). The catalytic and activation loops are marked as blue and green rectangles, respectively. Residues S157 and S162 from the activation loop are highlighted. Cyan and red squares denote the putative catalytically relevant residues: K37, E53, R128, D129, and D147. Alignment was performed using the ClustalW iterative algorithm [53]. The secondary structure of LmJean3 was predicted using the EMBOSS v6.5.7 tool [37] implemented in Geneious v9.1.7 [36]. The figure was designed using IBS v1.02 illustrator [101].

**Figure 4 biomolecules-09-00723-f004:**
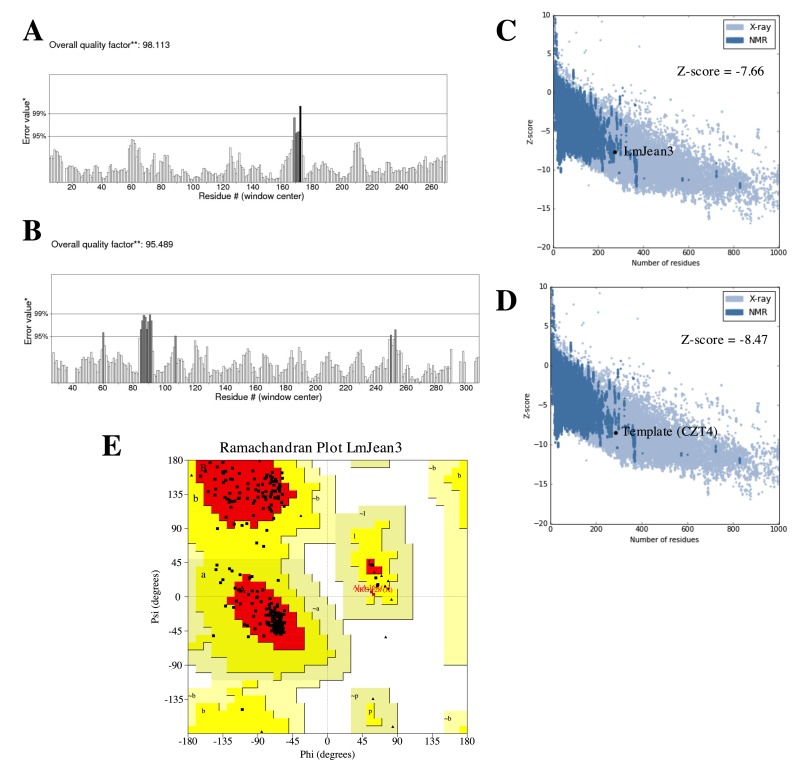
LmJean3 protein structure analysis and evaluation. Statistics of non-bonded interactions analyzed by ERRAT for (**A**) LmJean3 and (**B**) the template 4CZT_B. * On the error axis, two lines are drawn to indicate the confidence with which it is possible to reject regions that exceed that error value. ** Expressed as the percentage of the protein for which the calculated error value falls below the 95% rejection limit. (**C**,**D**) Comparative protein structure analysis (ProSA) Z-score evaluation for (**C**) LmJean3 and (**D**) 4CZT_B; (**E**) Ramachandran plot from the optimized structure for LmJean3. The homology model displayed 88.4% of the residues in favorable positions, and none was found in the disallowed regions.

**Figure 5 biomolecules-09-00723-f005:**
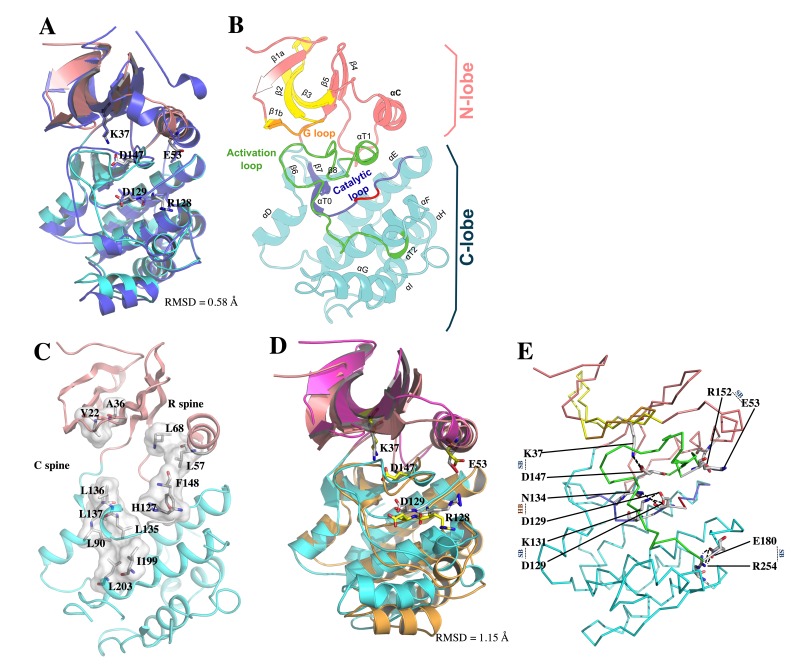
LmJean3 structure evaluation. (**A**) Superposition of the optimized LmJean3 homology model (pink N-lobe and cyan C-lobe) and 4CZT_B (purple) revealed LmJean3 catalytic residues K37, E53, R128, D129, and D147 aligned to their equivalents in the template (4CZT_B). The calculated RMSD was 1.87 Å; (**B**) predicted homology model for LmJean3 showing the ATP-binding region in yellow, glycine-rich loop in orange, activation loop in green, catalytic loop in blue, and catalytic residues R128 and D129 in red; (**C**) molecular surface representation of the catalytic and regulatory spines of LmJean3. The residues forming the spines are displayed as white sticks; (**D**) structure alignment of LmJean3 catalytically relevant residues (K37, E53, R128, D129, and D147) at the initial point of the simulation (magenta N-lobe and orange C-lobe) and the average calculated structure from the trajectory (pink N-lobe and cyan C-lobe). The catalytic residues from the initial frame are colored yellow, and their equivalents from the average structure are colored white. The calculated RMSD was 1.15 Å; (**E**) salt bridges (SB) and hydrogen bonds (HB), with persistence higher than 40% during the trajectory of the molecular dynamics’ simulation are shown in the average structure.

**Figure 6 biomolecules-09-00723-f006:**
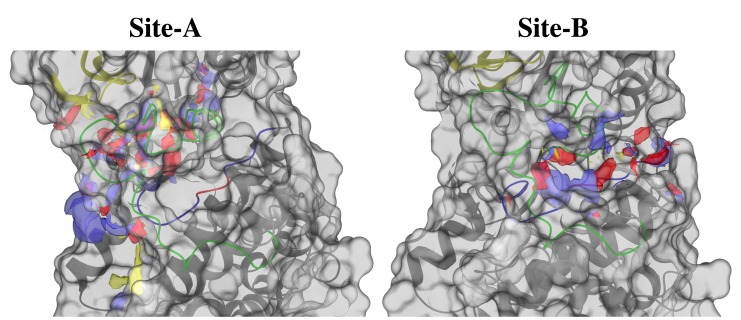
LmJean3 displays two promising binding sites for small-molecule, drug-like ligands. Hydrophobic (yellow), donor (blue), and acceptor (red) maps for (**A**) the ATP-binding pocket and the adjacent allosteric site and (**B**) the protein substrate binding site. SiteMap version 3.8 [82,83] was used to identify the binding sites, and Maestro Schrödinger Release 2017-2 to compose the figures.

**Figure 7 biomolecules-09-00723-f007:**
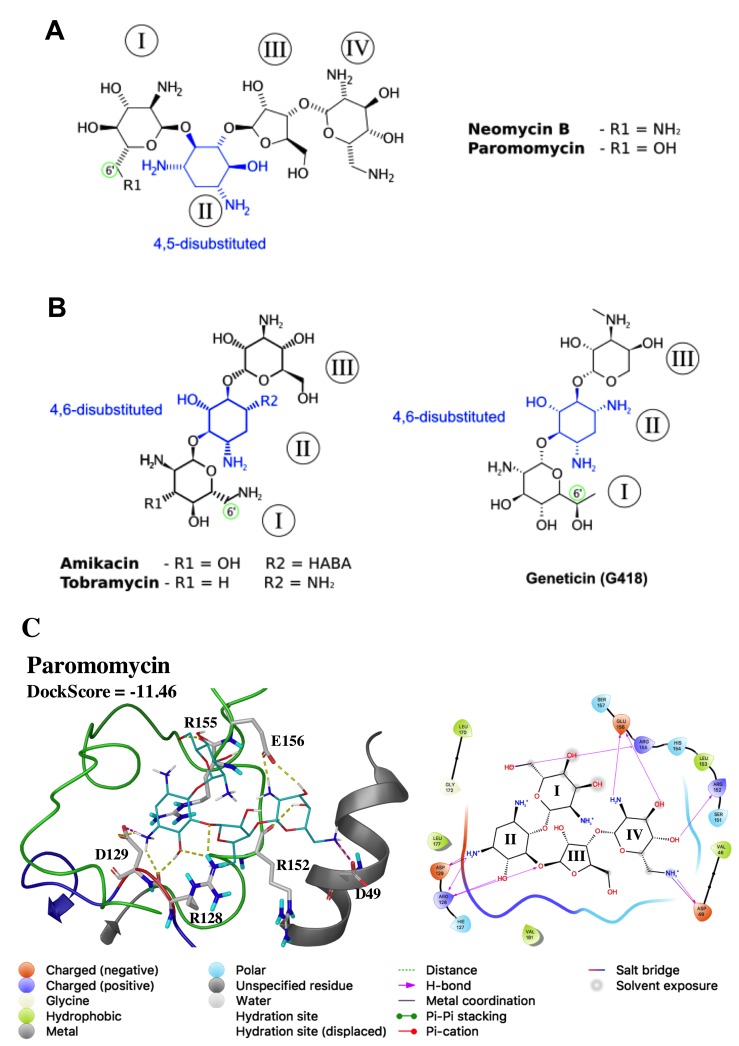
Representative aminoglycosides (AG) antibiotics of 4,5- and 4,6-disubstituted subclasses. Structure representation of (**A**) 4,5-disubsituted AGs (paromomycin and neomycin B) and (**B**) 4,6-disubstituted AGs (amikacin, tobramycin, and geneticin); HABA: α-hydroxy-γ-aminobutyric acid. The position 6′ from Ring I is highlighted by a green-colored circle; (**C**) predicted interactions of paromomycin with LmJean3 Site B. The 3D representation of the docking pose is shown on the left (distances up to 2.8 Å), and the ligand interaction diagram (distances up to 4 Å) on the right. Species in the ligand interaction diagram are colored by type: hydrophobic residues green, charged residues red (−) or violet (+), polar residues light blue and glycine light green. H-bond interactions with amino acids are highlighted as purple lines, salt bridges as blue- and red-colored lines. The figure was prepared using Maestro Schrödinger Release 2017-2.

**Figure 8 biomolecules-09-00723-f008:**
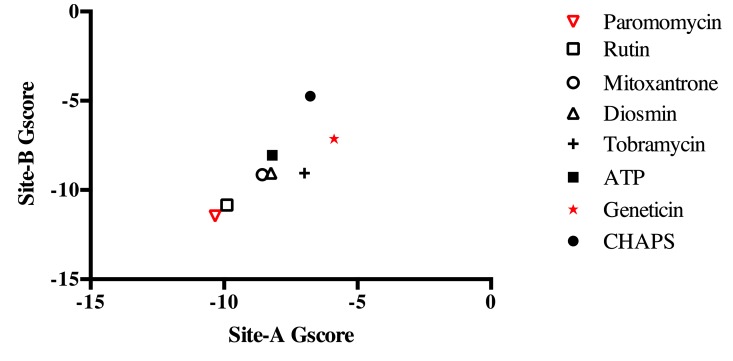
Site A and Site B docking scores. The top-ranked molecules resulting from docking to Site B were also employed to predict their docking scores to Site A. Additionally, CHAPS (present in the co-crystal structure) and ATP (co-factor) were also included in the analysis. The predicted GlideScores (kcal/mol) for each molecule are represented on the *X*-axis for Site A and on the *Y*-axis for Site B.

**Figure 9 biomolecules-09-00723-f009:**
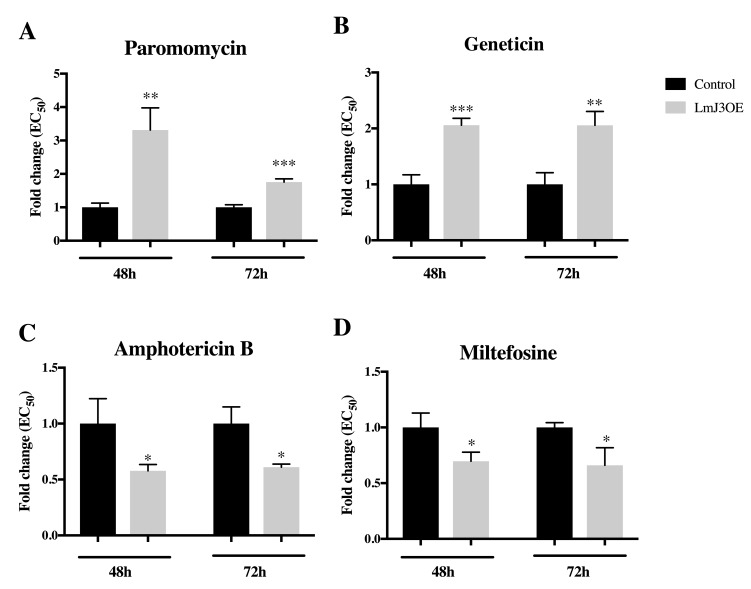
LmJean3-overexpression in *Leishmania* (LmJ3OE parasites) reduced the sensitivity to AG treatment after 48 h and 72 h. Leishmanicidal activity determination (EC_50_) by MTT assays of (**A**) paromomycin, (**B**) geneticin (G418), (**C**) amphotericin B, and (**D**) miltefosine in (mock) control and LmJ3OE parasites. Bars represent the normalized mean EC_50_ (± SD) from three independent experiments (* *p* < 0.05, ** *p* < 0.01, *** *p* < 0.001).

**Table 1 biomolecules-09-00723-t001:** Persistence of the SB and HB found within the kinase domain of LmJean3 during the molecular dynamics’ simulation (20 ns). See Appendix A for an exhaustive list of all the possible interactions found during the trajectory.

Donor	Acceptor	Bond Type	Persistence (>40%)
ARG254	GLU180	SB	100.00
LYS131	ASP129	SB	98.45
ARG152	GLU53	SB	85.66
LYS37	ASP147	SB	82.46
ARG128-Main	ASP192-Side	HB	60.49
LEU137-Main	GLY88-Main	HB	48.95
ASN134-Side	ASP129-Main	HB	40.61

**Table 2 biomolecules-09-00723-t002:** Pharmacokinetic parameters important for good oral bioavailability and docking results.

Rank	IUPAC Name	Chemical Name	MW	LogP	HBA (Lipinski)	HBD (Lipinski)	Violations (Lipinski)	Docking Score (kcal/mol)
1	(2S,3S,4R,5R,6R)-5-amino-2-(aminomethyl)-6-[(2R,3S,4R,5S)-5-[(1R,2R,3S,5R,6S)-3,5-diamino-2-[(2S,3R,4R,5S,6R)-3-amino-4,5-dihydroxy-6-(hydroxymethyl)oxan-2-yl]oxy-6-hydroxycyclohexyl]oxy-4-hydroxy-2-(hydroxymethyl)oxolan-3-yl]oxyoxane-3,4-diol	Paromomycin sulfate	615.6	−8.67	19	18	2	−11.46
2	(2R,3S,4R,5R,6R)-5-amino-2-(aminomethyl)-6-[(1R,2R,3S,4R,6S)-4,6-diamino-2-[(2S,3R,4S,5R)-4-[(2R,3R,4R,5S,6S)-3-amino-6-(aminomethyl)-4,5-dihydroxyoxan-2-yl]oxy-3-hydroxy-5-(hydroxymethyl)oxolan-2-yl]oxy-3-hydroxycyclohexyl]oxyoxane-3,4-diol	Neomycin sulfate stereoisomer A	614.6	−8.96	19	19	2	−10.93
3	2-(3,4-dihydroxyphenyl)-5,7-dihydroxy-3-[(2S,3R,4S,5S,6R)-3,4,5-trihydroxy-6-[[(2R,3R,4R,5R,6S)-3,4,5-trihydroxy-6-methyloxan-2-yl]oxymethyl]oxan-2-yl]oxychromen-4-one	Rutin	610.5	−1.16	16	10	2	−10.84
4	1,4-dihydroxy-5,8-bis[2-(2-hydroxyethylamino)ethylamino]anthracene-9,10-dione	Mitoxantrone	444.5	0.07	10	8	1	−9.14
5	(2S,3R,4S,5S,6R)-4-amino-2-[(1S,2S,3R,4S,6R)-4,6-diamino-3-[(2R,3R,5S,6R)-3-amino-6-(aminomethyl)-5-hydroxyoxan-2-yl]oxy-2-hydroxycyclohexyl]oxy-6-(hydroxymethyl)oxane-3,5-diol	Tobramycin sulfate	467.5	−6.86	14	15	2	−9.06
6	5-hydroxy-2-(3-hydroxy-4-methoxyphenyl)-7-[(2S,3R,4S,5S,6R)-3,4,5-trihydroxy-6-[[(2R,3R,4R,5R,6S)-3,4,5-trihydroxy-6-methyloxan-2-yl]oxymethyl]oxan-2-yl]oxychromen-4-one	Diosmin	608.5	−0.4	15	8	2	−9.05
7	(2S)-4-amino-N-[(1R,2S,3S,4R,5S)-5-amino-2-[(2S,3R,4S,5S,6R)-4-amino-3,5-dihydroxy-6-(hydroxymethyl)oxan-2-yl]oxy-4-[(2R,3R,4S,5S,6R)-6-(aminomethyl)-3,4,5-trihydroxyoxan-2-yl]oxy-3-hydroxycyclohexyl]-2-hydroxybutanamide	Amikacin stereoisomer	585.6	−8.43	18	17	2	−9.04

MW, molecular weight; LogP, logarithm of compound partition coefficient between n-octanol and water; HBA, number of hydrogen bond acceptors; HBD, number of hydrogen bond donors.

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
