# Peer review of "The Novel Serine/Threonine Protein Kinase LmjF.22.0810 from Leishmania major May Be Involved in the Resistance to Drugs such as Paromomycin"

_biomolecules, 2019, doi:10.3390/biom9110723_

Round 1

Reviewer 1 Report

To begin with, I want to make a research team to carry out this work. In my opinion, it is a very good quality work, which deserves to be published in this form.

Author Response

Dear Reviewer:

Thank you for reviewing our manuscript, for the time and effort in reading the document and writing your opinions and comments. Please find our point-by-point response to your comments as well as our new manuscript. We have adapted the document according to your suggestions. We sincerely hope that it is now ready for publication in Biomolecules.

Response to the Reviewer #1:

Reviewer #1

To begin with, I want to make a research team to carry out this work. In my opinion, it is a very good quality work, which deserves to be published in this form.

Thank you so much.

Reviewer 2 Report

This is a well written manuscript and the authors have used various methods (computational and experimentally) to support the homology model. 

It is thus recommended this manuscript be accepted after the following minor changes.

Page 4: "The 262-aminoacid" should be "The 262-amino acid" Page 4, the authors deleted the C-terminal 97 amino acids from the modeling using the concepts of "disordered region". Is there any functional significance being identified experimental (via mutational studies and the like) with C-terminal residues? If so, it would be problematic by simply deleting the C-terminal residues. In Figure 1, the authors listed the multiple sequence of query sequence to other template sequences. What's the sequence identity between the query sequence and the template sequence that was used to make the model? The author mentioned "high protein-sequence identity" (page 2), but fell short of giving the exact number of sequence identity. Page 7 mentioned > 85%. However, by looking at Figure 1B, the MSA does not support the statement that there's >85% sequence identify. Page 13, the authors stated that 88.4% residues in the favorable region and none in teh dissallowed regions. However, Fig. 4E does show that at least one residue in the lower right corner, and one in the lower left corner are in the disallowed region. Can the double check the Ramachandran plot analysis? Font for Figure 5 A, B, C, D is too small to read in 100% settings. Please use larger font.

Author Response

Thank you.

Reviewer 3 Report

Minor comments:

Line 221. Is there any known effect of antibiotic cocktail on Leishmania major parasite?

Vacas and colleagues describe the identification of a novel serine/threonine-protein kinase (LmjF.22.0810) from Leishmania major. After cloning and overexpressing LmjF.22.0810 in L. major, parasites exhibited less sensitivity to paromomycin and geneticin. The authors predict that aminoglycosides preferentially bind to the phosphate site of the protein and may explain the observed drug resistance in transgenic parasites. Two druggable sites on the protein were also identified in the study.

The study is exciting and within the scope of the journal. The results are adequately supported by different experiments and discussion is well written. I would recommend acceptance of the manuscript in its current form.

Author Response

Thank you.
